# Polygenic scores, diet quality, and type 2 diabetes risk: An observational study among 35,759 adults from 3 US cohorts

Jordi Merino[1,2,3]☯*, Marta Guasch-Ferré[4,5]☯, Jun Li[4,6]☯, Wonil Chung[6,7], Yang Hu[4], Baoshan Ma[6,7], Yanping Li[4], Jae H. Kang[5], Peter Kraft[6,8,9], Liming Liang[6,8,9], Qi Sun[4,5,6], Paul W. Franks[4,10], JoAnn E. Manson[4,5,11], Walter C. Willet[4,5,6], Jose C. Florez[1,2,3‡], Frank B. Hu[4,5,6‡]

1 Diabetes Unit and Center for Genomic Medicine, Massachusetts General Hospital, Boston, Massachusetts, United States of America, 2 Program in Medical and Population Genetics, Broad Institute, Cambridge, Massachusetts, United States of America, 3 Department of Medicine, Harvard Medical School, Boston, Massachusetts, United States of America, 4 Department of Nutrition, Harvard TH Chan School of Public Health, Boston, Massachusetts, United States of America, 5 Channing Division of Network Medicine, Department of Medicine, Brigham and Women's Hospital and Harvard Medical School, Boston, Massachusetts, United States of America, 6 Department of Epidemiology, Harvard TH Chan School of Public Health, Boston, Massachusetts, United States of America, 7 College of Information Science and Technology, Dalian Maritime University, Dalian, Liaoning Province, China, 8 Program in Genetic Epidemiology and Statistical Genetics, Harvard TH Chan School of Public Health, Boston, Massachusetts, United States of America, 9 Department of Biostatistics, Harvard TH Chan School of Public Health, Boston, Massachusetts, United States of America, 10 Department of Clinical Sciences, Lund University, Malmö, Sweden, 11 Division of Preventive Medicine, Department of Medicine, Brigham and Women's Hospital and Harvard Medical School, Boston, Massachusetts, United States of America

☯ These authors contributed equally to this work.
‡ JCF and FBH also contributed equally to this work.
* jmerino@mgh.harvard.edu

**Data Availability Statement:** The data underlying the generation of the global polygenic score for type 2 diabetes are available from the UK Biobank project site, subject to registration and application

## Abstract

### Background

Both genetic and lifestyle factors contribute to the risk of type 2 diabetes, but the extent to which there is a synergistic effect of the 2 factors is unclear. The aim of this study was to examine the joint associations of genetic risk and diet quality with incident type 2 diabetes.

### Methods and findings

We analyzed data from 35,759 men and women in the United States participating in the Nurses' Health Study (NHS) I (1986 to 2016) and II (1991 to 2017) and the Health Professionals Follow-up Study (HPFS; 1986 to 2016) with available genetic data and who did not have diabetes, cardiovascular disease, or cancer at baseline. Genetic risk was characterized using both a global polygenic score capturing overall genetic risk and pathway-specific polygenic scores denoting distinct pathophysiological mechanisms. Diet quality was assessed using the Alternate Healthy Eating Index (AHEI). Cox models were used to calculate hazard ratios (HRs) for type 2 diabetes after adjusting for potential confounders. With over 902,386 person-years of follow-up, 4,433 participants were diagnosed with type 2 diabetes. The relative risk of type 2 diabetes was 1.29 (95% confidence interval [CI] 1.25, 1.32;

process. Further details can be found at https://www.ukbiobank.ac.uk. This research was conducted under UK Biobank application no. 45052. Code to run the genome-wide association analysis for type 2 diabetes and generate the global polygenic score has been uploaded to GitHub (https://github.com/lab319/ps-diet-t2d). Information including the procedures to obtain and access the data and codes used in this study in the Nurses' Health Study I and II, and the Health Professionals Follow-Up Study is described at http://www.nurseshealthstudy.org/researchers for the Nurses' Health Study (contact: nhsaccess@channing.harvard.edu) or https://www.hsph.harvard.edu/hpfs/ for the Health Professionals Follow-up Study (contact: hpfs@hsph.harvard.edu). The scripts to analyze NHS/HPFS data presented in this manuscript are open and widely available once access to the system is granted.

**Funding:** This study was funded by research grants from the National Institutes of Health CA186107 (J.E.M.), CA176726 (W.C.W.), CA167552 (W.C.W.), HL034594 (J.E.M.), HL035464 (Q.S.), EY015473 (F.B.H.), DK112940 (F.B.H.), DK120870 (F.B.H.), DK40561 (J.M), and DK110550 (J.C.F.), the American Diabetes Association 1-18-PMF-029 (M.G-F.), and the National Natural Science Foundation of China 61471078 (B.M.). The funders had no role in study design, data collection and analysis, decision to publish, or preparation of the manuscript.

**Competing interests:** I have read the journal's policy and the authors of this manuscript have the following competing interests: Y.L. have received research supports from the California Walnut Commission and Swiss Reinsurance Company, which were both unrelated with current research project. P.W.F. has received honoraria from Lilly Inc, Novo Nordisk A/S, and Zoe Global Ltd, with stock options. All other authors declare no competing interests.

**Abbreviations:** AHEI, Alternate Healthy Eating Index; BMI, body mass index; CI, confidence interval; DASH, Dietary Approaches to Stop Hypertension; GWAS, genome-wide association study; HPFS, Health Professionals Follow-up Study; HR, hazard ratio; IQR, interquartile range; NHS, Nurses' Health Study; RERI, relative excess risk due to interaction; SD, standard deviation; STROBE, Strengthening the Reporting of Observational Studies in Epidemiology.

$P < 0.001$) per standard deviation (SD) increase in global polygenic score and 1.13 (1.09, 1.17; $P < 0.001$) per 10-unit decrease in AHEI. Irrespective of genetic risk, low diet quality, as compared to high diet quality, was associated with approximately 30% increased risk of type 2 diabetes ($P_{interaction} = 0.69$). The joint association of low diet quality and increased genetic risk was similar to the sum of the risk associated with each factor alone ($P_{interaction} = 0.30$). Limitations of this study include the self-report of diet information and possible bias resulting from inclusion of highly educated participants with available genetic data.

## Conclusions

These data provide evidence for the independent associations of genetic risk and diet quality with incident type 2 diabetes and suggest that a healthy diet is associated with lower diabetes risk across all levels of genetic risk.

## Author summary

### Why was this study done?

- Both genetic and lifestyle factors contribute to individual-level risk of type 2 diabetes.
- While previous studies have shown that adherence to a healthy lifestyle is associated with reduced risk of type 2 diabetes regardless of genetic risk, the partial characterization of genetic risk and the predominant assessment of interactions on the multiplicative scale might have prevented previous studies from identifying genetic profiles interacting with dietary exposures.
- Therefore, understanding how genetic risk and diet quality contribute to the development of type 2 diabetes is important to support evidence-based preventive interventions.

### What did the researchers do and find?

- In 3 cohort studies involving 35,759 men and women in the US, we used novel polygenic scores for type 2 diabetes to systematically evaluate the presence of additive and multiplicative interactions between genetic risk and diet quality on the development of type 2 diabetes.
- We found that both low diet quality and increased overall or pathway-specific genetic risk were independently associated with higher risk of type 2 diabetes.
- We documented that within any genetic risk category, high diet quality, as compared to low diet quality, was associated with a nearly 30% lower risk of type 2 diabetes.
- Further, we showed that the risk of type 2 diabetes attributed to the combination of increased genetic risk and low diet quality was similar to the sum of the risks associated with each factor alone.

**What do these findings mean?**

- Results from this study suggest that consuming a healthier diet is associated with a lower risk of type 2 diabetes regardless of genetic risk.

- Our results underscore the value of genetic risk assessment to identify individuals at increased disease risk and their potential for risk stratification and surveillance.

- Such knowledge can serve to inform and design future strategies to advance the prevention of type 2 diabetes.

## Introduction

The burden of type 2 diabetes is not equally distributed, as susceptibility to environmental factors varies between and within human populations [1]. This observation has led many to presume that dietary and lifestyle factors may yield different effects depending on inherited genetic susceptibility, a concept often referred to as "gene × lifestyle interaction" [2–4]. To date, some studies have attempted to identify genotypes interacting with lifestyle factors on the development of type 2 diabetes, but these studies have consistently demonstrated that adherence to healthy dietary or lifestyle recommendations is associated with a lower burden of type 2 diabetes regardless of genetic risk [5–11]. Partial characterization of genetic risk, often based on polygenic scores that included a limited number of variants, the predominant assessment for interactions on the multiplicative scale alone, or the use of a single time point dietary exposure assessment and limited follow-up might have prevented previous studies from identifying genotypes interacting with lifestyle or dietary exposures.

Recent genetic discoveries and improved computational algorithms offer an unprecedented opportunity to better characterize type 2 diabetes genetic risk [12,13]. It is now possible to clump thousands of genetic variants with marginal effects into a "global" polygenic score with considerable impact on disease risk [12]. In addition, it is possible to capture the etiological heterogeneity that characterizes type 2 diabetes by generating "pathway-specific" polygenic scores with variants that share increased type 2 diabetes risk through specific pathophysiological processes such as impaired insulin secretion or different forms of insulin resistance [13]. The extent to which this knowledge is useful for identifying individuals more susceptible than others to an unhealthy diet is unknown.

Here, we analyzed longitudinal data for 35,759 participants in 3 cohorts to investigate how genetic risk and diet quality contribute to the risk of type 2 diabetes.

## Methods

### Study design and population

We used data collected from 3 prospective cohort studies in the US including participants in the Nurses' Health Study (NHS), the Health Professionals Follow-up Study (HPFS), and the NHS II [14,15]. The NHS was established in 1976 when 121,700 female registered nurses aged 30 to 55 were recruited [14]. The HPFS began in 1986 and enrolled 51,529 male health professionals aged 40 to 75 years [15]. The NHS II cohort was initiated in 1989 and included 116,340 women aged 25 to 42 years [14]. The study baseline was set at 1986 for the NHS and HPFS and

1991 for the NHS II, which was when participants first completed a questionnaire on their medical history, diet, and lifestyle characteristics.

Multiple genome-wide association studies (GWASs) have been conducted within the NHS, NHS II, and HPFS nested cohorts to investigate genetic susceptibility to 12 complex diseases [16]. Participants for genetic determinations were selected to represent a representative sample of the original sample. Demographic characteristics and health status of participants with genetic information were generally similar to those without genetic information (S1 Table). Genotype, imputation, and quality control of genome-wide genetic data have been harmonized across nested cohorts and detailed elsewhere [16]. After quality control, genome-wide genetic data were available for 42,437 individuals. We excluded participants diagnosed with diagnosis of type 2 diabetes (n = 3,200), cardiovascular disease (including nonfatal myocardial infarction, fatal coronary heart disease, and fatal and nonfatal stroke, n = 613), or cancer at baseline (n = 721), those who had an unusual total energy intake at baseline (<800 kcal or >4,200 kcal/day in men and <500 or >3,500 kcal/day in women, n = 581), and those who completed only the baseline questionnaire (n = 1,563). After these exclusions, 14,454 participants in the NHS, 9,417 participants in the HPFS, and 11,888 participants in the NHS II were included in this analysis. The study protocol was approved by the human research committee of Brigham and Women's Hospital and the Harvard TH Chan School of Public Health.

## Ascertainment of type 2 diabetes

Cases of type 2 diabetes were identified by biennially mailed questionnaires and confirmed by a validated supplementary questionnaire regarding symptoms, diagnostic laboratory test results, and hypoglycemic therapy. For cases diagnosed before 1998, type 2 diabetes was documented if participants met at least 1 of the following National Diabetes Data Group criteria [17]: (a) raised glycemia (fasting plasma glucose $\geq$ 7.8 mmol/l, random plasma glucose $\geq$ 11.1 mmol/l, or plasma glucose $\geq$ 11.1 mmol/l after an oral glucose load) and at least 1 symptom related to diabetes (excessive thirst, hunger, polyuria, or weight loss); (b) no symptoms, but elevated glucose concentrations on 2 occasions; and (c) treatment with insulin or other hypoglycemic medication [10]. From 1998 onward, the cutoff point for elevated fasting plasma glucose concentrations was lowered to 7.0 mmol/l according to the American Diabetes Association criteria [18]. We also considered a HbA1c concentration $\geq$6.5% criteria for confirming type 2 diabetes cases identified after January 2010 [19]. Validation studies in subsamples of the NHS revealed the validity of using the supplementary questionnaires to adjudicate type 2 diabetes diagnosis, showing that more than 97% of participants with self-reported type 2 diabetes were reconfirmed through medical record review [20].

## Type 2 diabetes polygenic scores

We generated a global polygenic score for type 2 diabetes that captures overall genetic burden using external data from UK Biobank (S1 Text). The rationale to use external data from UK Biobank was to avoid sample overlap with a previous publicly available global polygenic score for type 2 diabetes [12]. In brief, we selected a random UK Biobank sample (n = 391,147 participants, 17,403 type 2 diabetes cases) and conducted a genome-wide association analysis for type 2 diabetes using linear mixed models implemented in BOLT-LMM [21]. Next, estimated effect sizes were reweighted and clumped using LDPred [22]. The predictive performance of the global polygenic score including approximately 850,000 independent genetic variants was tested in the remaining set of UK Biobank participants (n = 20,000 participants, 893 type 2 diabetes cases) and then applied to our study population. To calculate individual scores in our study population, each variant was coded with the expected number of associated alleles and

weighted by its relative effect size on type 2 diabetes. The scores, which included the same number of genetic variants in each cohort, were then standardized.

To generate pathway-specific polygenic scores, we used data from a previous study aimed at grouping known type 2 diabetes loci based on shared physiological similarities [13]. Genetic variants to compute these polygenic scores and their respective weights are detailed in S2 Table. These pathway-specific polygenic scores capture biological processes relevant to diabetes pathophysiology including impaired insulin secretion (one polygenic score for beta-cell dysfunction and another for proinsulin synthesis) and increased insulin resistance (polygenic scores related to obesity-mediated insulin resistance, body fat distribution, and lipid/hepatic metabolism). Allocation of type 2 diabetes variants to each polygenic score is supported by tissue-specific patterns of chromatin accessibility, histone modification, and transcriptional regulation [13], indicating that the mechanistic basis of these polygenic scores is robust even though these variants may have pleiotropic effects. The significance of pathway-specific polygenic scores has been shown in previous studies indicating that individuals enriched for genetic variants defining each of the intermediate diabetogenic processes exhibited the predicted score–associated phenotypes [13,23]. The scores were generated by multiplying a variant's genotype dosage by its respective weight and then standardized. Polygenic scores were standardized to allow comparisons across scores computed in this study with different number of genetic variants.

## Assessment of diet quality

Diet quality was assessed using diet information obtained from a validated 131-item semi-quantitative food frequency questionnaire administered at baseline and every 4 years thereafter. To quantify overall diet quality, we calculated the Alternate Healthy Eating Index (AHEI) using food components and scoring criteria that have been described previously [24]. The AHEI score is based on 11 foods and nutrients, emphasizing higher intake of fruits, whole grains, vegetables (excluding potatoes), nuts and legumes, polyunsaturated fatty acids, and long chain (n-3) fatty acids; moderate intake of alcohol; and lower intake of red and processed meats, sugar sweetened drinks and fruit juice, sodium, and *trans*-fat. Each component was scored from 0 (unhealthiest) to 10 (healthiest) points, with intermediate values scored proportionally, and all component scores were summed to obtain a total score ranging from 0 (lowest diet quality) to 110 (highest diet quality) points.

As an additional method to quantify diet quality, we used the Dietary Approaches to Stop Hypertension (DASH) score [25]. The DASH score was based on the DASH-style diet, which includes information from 8 foods and nutrients. Each component was scored from 1 to 5 points according to fifths of intake, with 5 being the best score for higher intake of fruits, whole grains, vegetables, nuts and legumes, and low-fat dairy products and for lower intake of red and processed meats, sugar sweetened drinks, and sodium. The total score ranged from 8 (lowest diet quality) to 40 (highest diet quality) points.

## Assessment of covariates

Covariates were ascertained every 2 years with the use of questionnaire that obtained updated information on occurrence of diseases and many lifestyles and personal risk factors, including age, family history of diabetes, history of hypertension, history of hypercholesterolemia, body mass index (BMI), menopausal status and postmenopausal hormone use in women, smoking status, physical activity, total energy intake, and alcohol intake. Baseline history of hypertension and hypercholesterolemia were determined through self-reporting. BMI was calculated as weight in kilograms divided by the square of the height in meters. Physical activity was

repeatedly assessed using validated questionnaire on time spent on recreational activities. We used principal component analysis in each cohort to generate ancestry-derived principal components.

## Statistical analysis

We elaborated a prespecified protocol including definitions of exposures, outcomes and covariates, and statistical analysis plan prior to data analysis (S2 Text). We summarized continuous measurements by using means (standard deviation, SD) or medians (interquartile range, IQR) and present categorical observations as frequency and percentages. Correlations between diet and polygenic scores were assessed using Pearson correlation test. To better capture longitudinal trajectories of diet quality, we calculated and used cumulative averages of diet quality. To generate cumulative averages, we continually updated diet throughout duration of follow up. Because the proportion of missing values of covariates for individuals with genetic data was below 5%, participants with missing covariate information were excluded from the analysis.

Person-time for each participant was calculated from the return of the baseline questionnaire to the diagnosis of type 2 diabetes, death, loss to follow-up, or the end of the follow-up period (2016 for the NHS and the HPFS and 2017 for the NHS II), whichever came first. We used multivariable Cox proportional hazards models to calculate hazard ratios (HRs) and 95% confidence intervals (CIs) for type 2 diabetes after exploring that the proportional hazards assumption was not violated. The proportionality of hazards assumption was assessed using the Schoenfeld residuals. We modeled polygenic scores and diet quality as continuous variables. We have used as time-varying variables in the models, the variables that change over time including age, history of hypertension, history of hypercholesterolemia, menopausal status, BMI, smoking status, physical activity, and total energy intake. Family history of type 2 diabetes and ancestry-derived principal components were considered time-fixed variables at baseline. For the variables that were updated throughout the follow-up, the questionnaire year determined the time point. Cox regression models were stratified by age (in months, continuous) and adjusted for ancestry-derived principal components (1 to 4) (crude model). The multivariable-adjusted model was further adjusted for family history of diabetes (yes or no), history of hypertension (yes or no), history of hypercholesterolemia (yes or no), menopausal status (premenopausal or postmenopausal [never, past, or current menopausal hormone use], women only), BMI (quintiles of kg/m$^2$), smoking status (current, former, and never), physical activity (quintiles of MET hours/week), and total energy intake (quintiles of total caloric intake/day). Because BMI could be on the causal pathway between diet quality and type 2 diabetes risk, we also conducted separate models without adjusting for BMI.

We conducted analyses stratified by genetic risk category (low, intermediate, and high based on thirds of genetic risk distribution) to assess the association between diet quality and type 2 diabetes risk. We also cross-classified participants according to categories of genetic risk and diet quality (9 categories based on thirds of genetic risk and diet quality score, with low genetic risk and high diet quality as reference) and conducted joint analyses to investigate the combined association of genetic risk and diet quality with the risk of type 2 diabetes.

We evaluated whether the associations between diet quality and type 2 diabetes risk differed based on genetic susceptibility by using additive and multiplicative interaction analyses [26,27]. Power calculations for interaction analyses were conducted to determine the minimum detectable interaction on the risk ratio scale [28]. The available sample gave us 80% statistical power at α 0.05 to detect an additive and multiplicative interaction effect size ≥1.04 and ≥1.10, respectively. We tested for multiplicative interactions using the log-likelihood ratio test to compare the goodness of fit of a multivariable-adjusted model with and without the

cross-product interaction term [27]. For additive interaction analyses, we considered genetic risk as a continuous variable and used a binary categorical variable for diet quality based on the median of the diet quality score in each cohort. We assessed the relative excess risk due to interaction (RERI) as an index of additive interaction [26] and further examined the decomposition of the joint effect, which is the proportion of risk due to genetic risk alone, to diet quality alone, and to their interaction [26]. CIs for each of the interaction measures were calculated using the delta method described by Hosmer and Lemeshow [29].

We conducted secondary analyses to investigate the consistency of our results. First, we used the DASH score as an additional method to quantify diet quality. For these analyses, we further adjusted our multivariable models for alcohol intake. We tested for additive and multiplicative interactions using the DASH score. Second, we conducted 3-way interaction analyses to examine whether BMI modified the joint association between increased genetic risk and low diet quality on type 2 diabetes risk.

All analyses were performed separately for each cohort and then were pooled with the use of inverse variance weighted, fixed-effects meta-analysis. Results were also combined using random-effects meta-analysis. The heterogeneity index ($I^2$) was used to assess heterogeneity. All $P$ values presented were 2 sided, with statistical significance determined by the Bonferroni corrected threshold of significance <0.007 (0.05/7 exposures). Data were analyzed with the use of SAS software, version 9.3 (SAS Institute) and R software, version 4.0.3 (R Foundation). This manuscript is reported as per the Strengthening the Reporting of Observational Studies in Epidemiology (STROBE) guideline (S3 Text) [30].

## Results

Baseline characteristics of the 35,759 participants included in this study are shown in Table 1. The mean baseline age of NHS participants was 53 years old, 54 in HPFS, and 37 in NHS II. Most of them were of European descent without major chronic diseases at baseline. Mean BMI ranged from 24.3 kg/m$^2$ in NHS to 25.5 kg/m$^2$ in NHS II. At baseline, mean AHEI score ranged from 48.9 in NHS II to 52.6 in HPFS. The study sample was representative of each original study population with no major differences in clinical, demographic, and lifestyle characteristics (S1 Table). A total of 4,433 participants developed type 2 diabetes during 902,387 person-years of follow-up ($n$ = 2,204 (15.2%) in NHS, $n$ = 1,285 (13.6%) in HPFS, and $n$ = 944 (7.9%) in NHS II).

### Associations between polygenic scores and type 2 diabetes incidence

The polygenic scores were normally distributed (S1–S3 Figs). The age-adjusted HR for type 2 diabetes was 1.42 (95% CI 1.38, 1.46; $I^2$ = 93.2%; $P$ < 0.001) per 1 SD increase in the global polygenic score (S3 Table). In fully adjusted models, the global polygenic score was associated with higher risk of type 2 diabetes with an HR of 1.29 (95% CI 1.25, 1.33; $I^2$ = 88.4%; $P$ < 0.001; per SD increase; Fig 1). When analyzed in each cohort separately, the multivariable-adjusted HR for type 2 diabetes was 1.26 (95% CI 1.20, 1.31; $P$ < 0.001) in NHS, 1.23 (95% CI 1.16, 1.31; $P$ < 0.001) in HPFS, and 1.46 (95% CI 1.37, 1.56; $P$ < 0.001) in NHS II. When pathway-specific polygenic scores were used to characterize genetic risk, there were consistent associations between increased genetic risk and type 2 diabetes risk. The crude estimates for pathway-specific polygenic scores are presented in S3 Table. The multivariable-adjusted HRs per 1 SD increase in pathway-specific polygenic scores ranged from 1.26 (95% CI 1.22, 1.30; $I^2$ = 55.5%; $P$ < 0.001) for the beta-cell dysfunction polygenic score to 1.09 (95% CI 1.05, 1.12; $I^2$ = 49.1%; $P$ < 0.001) for the obesity-mediated insulin resistance polygenic score (Fig 1).

**Table 1. Baseline characteristics of the 35,759 US men and women in the NHS I, the HPFS, and the NHS II.**

| Characteristic | NHS ($n$ = 14,454) | HPFS ($n$ = 9,417) | NHS II ($n$ = 11,888) |
|---|---|---|---|
| Person-y$^\phi$ | 366,719 | 239,296 | 296,371 |
| Age, years | 53 (7) | 54 (9) | 37 (4) |
| Self-reported race/ethnicity$^\epsilon$ | | | |
| White, $n$ (%) | 14,416 (99.7) | 9,267 (98.4) | 11,845 (99.6) |
| Other, $n$ (%) | 38 (0.3) | 150 (1.6) | 43 (0.4) |
| **Clinical history** | | | |
| Hypertension, $n$ (%) | 2,193 (15.2) | 1,833 (19.5) | 345 (2.9) |
| Dyslipidemia, $n$ (%) | 1,133 (7.8) | 1,117 (11.9) | 1,173 (9.9) |
| Family history of diabetes, $n$ (%) | 4,323 (29.9) | 2,695 (28.6) | 4,294 (36.1) |
| Hormone use, $n$ (%) | | | |
| Premenopausal | 5,742 (39.7) | — | 9,617 (80.9) |
| Postmenopausal, never | 3,870 (26.8) | — | 1,340 (11.3) |
| Postmenopausal, current | 2,535 (17.5) | — | 819 (6.9) |
| Postmenopausal, previous | 1,902 (13.2) | — | — |
| **Lifestyle habits** | | | |
| Current smoker, $n$ (%) | 2,479 (17.2) | 728 (7.7) | 1,245 (10.5) |
| BMI, kg/m$^2$ | 25.2 (4.6) | 25.5 (3.1) | 24.3 (5) |
| Physical activity, MET-h/wk, median (IQR) | 14.3 (2.9 to 19.3) | 19.1 (4.3 to 25.5) | 20.2 (5.2 to 26.0) |
| Total energy intake, kcal/day, mean (SD) | 1,781 (520) | 1,988 (557) | 1,802 (535) |
| AHEI score, mean (SD)* | 52.1 (11.3) | 52.6 (11.7) | 48.9 (11) |
| Alcohol intake, g/day, median (IQR) | 6.5 (0 to 8.6) | 12.2 (1.1 to 16.1) | 3.2 (0 to 3.5) |

Values are means (SD) or medians (IQR) for continuous variables and numbers and percentages are for categorical variables. The study baseline was set at 1986 for the NHS I and the HPFS and 1991 for the NHS II.

MET denotes metabolic equivalent tasks.

$^\phi$Person-years are based on the analysis for type 2 diabetes.

$^\epsilon$Race was self-reported by the participants. Non-Hispanic white (southern European/Mediterranean, Scandinavian, and other European ancestry) and Hispanic were categorized into "White," while Black, Asian, American Indian, or Hawaiian were categorized into "Other." Ancestry-derived principal components were used to adjust multivariable models.

*Scores on the AHEI range from 0 to 110, with higher scores indicating a healthy diet.

AHEI, Alternate Healthy Eating Index; BMI, body mass index; HPFS, Health Professionals Follow-up Study; IQR, interquartile range; NHS, Nurses' Health Study; SD, standard deviation.

The correlations between polygenic scores included in this study were modest ($r^2$ ranging from 0.27 to 0.07), supporting the notion that they capture different axes of genetic predisposition (S4 Table). The association between polygenic scores and type 2 diabetes risk was consistent in models without adjusting for BMI (S3 Table) or when random-effects meta-analyses were used to combine cohort estimates (S5 Table).

## Interplay between diet quality and genetic risk on the development of type 2 diabetes

The risk of type 2 diabetes per 10-unit decrease AHEI score was 1.13 (95% CI 1.09, 1.17; $I^2$ = 58.6%; $P <$ 0.001) after adjusting for potential confounders (S4 Fig). When analyzed in each cohort separately, the risk of type 2 diabetes per 10-unit decrease in AHEI score was 1.11 (95% CI 1.06, 1.17; $P <$ 0.001) in the NHS, 1.20 (95% CI 1.12, 1.28; $P <$ 0.001) in the HPFS, and 1.08 (95% CI 1.00, 1.16; $P =$ 0.048) in the NHS II. The association between diet quality and type 2 diabetes risk was consistent in secondary analyses using the DASH score (pooled HR 1.13, 95

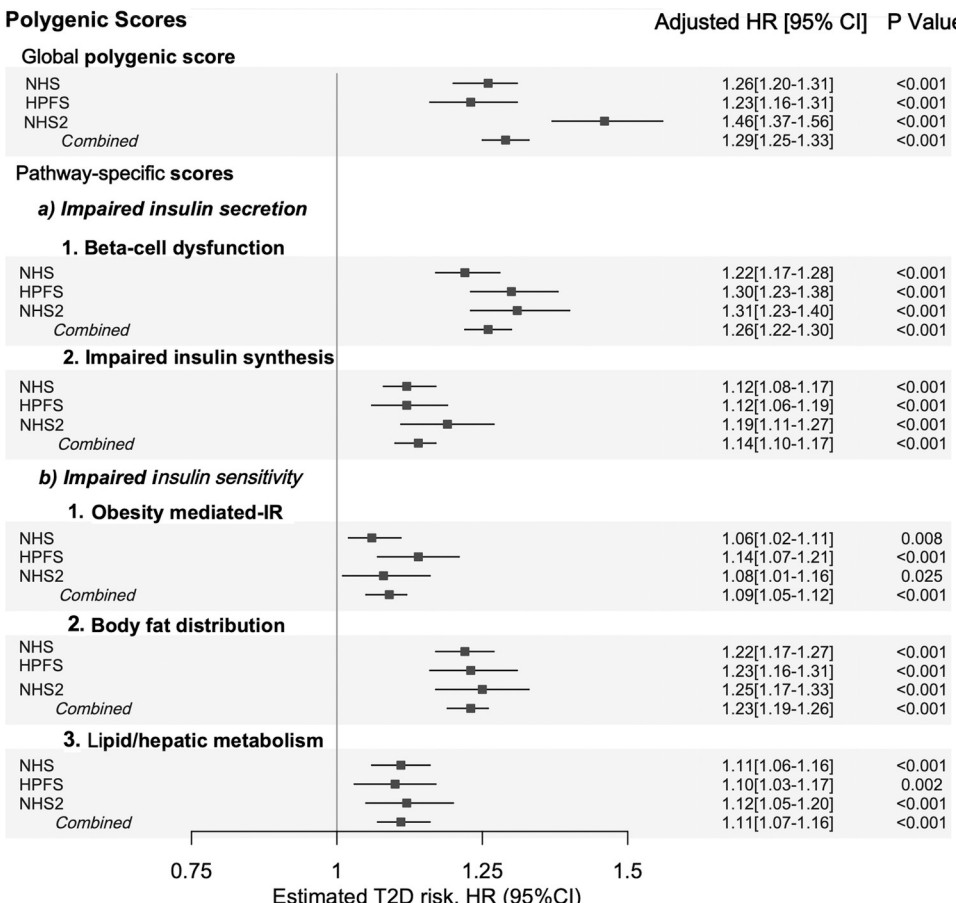

**Fig 1. Risk of incident type 2 diabetes associated with genetic risk.** Shown are adjusted HRs and 95% CI of the estimate for type 2 diabetes risk per SD increase in polygenic scores. Estimates are presented for each of the 3 prospective cohorts separately and in a combined analysis. Polygenic scores included in this study are described in the methods section. Cox proportional hazards models were stratified by age and adjusted for ancestry-derived principal components, family history of diabetes, history of hypertension, history of hypercholesterolemia, menopausal status (women only), BMI, smoking status, physical activity, and total energy intake. Fixed-effects inverse variance weighted meta-analysis was used to combine cohort-specific results. BMI, body mass index; CI, confidence interval; HPFS, Health Professionals Follow-up Study; HR, hazard ratio; IR, insulin resistance; NHS, Nurses' Health Study; NHS2, Nurses' Health Study II; SD, standard deviation; T2D, type 2 diabetes.

CI 1.09, 1.18; $I^2$ = 81.7%; $P < 0.001$; per 5-unit lower; S5 Fig). The correlation between the AHEI and the DASH score was high ($r^2 > 0.6$ in all 3 cohorts, $P < 0.001$).

When analyzed within each category of genetic risk, defined using the global polygenic score, low diet quality was consistently associated with higher type 2 diabetes risk (Table 2). The unadjusted HR for type 2 diabetes when compared individuals in the lowest category of the diet quality score to those at the highest category was 1.74 (95% CI 1.48, 2.05; $P < 0.001$) among participants at low genetic risk, 1.84 (95% CI 1.59, 2.13; $P < 0.001$) among participants at intermediate genetic risk, and 1.72 (95% CI 1.53, 1.93; $P < 0.001$) among participants at high genetic risk. In the multivariable model, low diet quality, as compared to high diet quality, was associated with higher risk of type 2 diabetes with an adjusted HR of 1.31 (95% CI 1.09, 1.58; $P = 0.001$) among participants at low genetic risk, 1.39 (95% CI 1.19, 1.63; $P < 0.001$) among those at intermediate genetic risk, and 1.29 (95% CI 1.14, 1.46; $P < 0.001$) among those at high genetic risk. Findings were consistent in models without adjusting for BMI (Table 2).

**Table 2. Association between diet quality and type 2 diabetes risk according to categories of genetic risk.**

| Subgroup | HR (95% CI) | *P* value |
|---|---|---|
| **Low genetic risk** | | |
| Crude model | 1.74 (1.48, 2.05) | <0.001 |
| Multivariable-adjusted model | 1.31 (1.11, 1.54) | 0.001 |
| Multivariable-adjusted without BMI | 1.41 (1.17, 1.69) | <0.001 |
| **Intermediate genetic risk** | | |
| Crude model | 1.84 (1.59, 2.13) | <0.001 |
| Multivariable-adjusted model | 1.39 (1.19, 1.63) | <0.001 |
| Multivariable-adjusted without BMI | 1.50 (1.29, 1.75) | <0.001 |
| **High genetic risk** | | |
| Crude model | 1.72 (1.53, 1.93) | <0.001 |
| Multivariable-adjusted model | 1.29 (1.14, 1.46) | <0.001 |
| Multivariable-adjusted without BMI | 1.38 (1.22, 1.56) | <0.001 |

HRs and 95% CI for type 2 diabetes risk for low versus high diet quality according to genetic risk categories. Cox proportional hazards models were stratified by age (in months, continuous) and adjusted for ancestry-derived principal components (1 to 4) (crude model). Multivariable-adjusted model was further adjusted for time-dependent confounders including family history of diabetes (not time-dependent, yes or no), hypertension (yes or no), hypercholesterolemia (yes or no), menopausal status (premenopausal or postmenopausal [never, past, or current menopausal hormone use], women only), BMI (quintiles of $kg/m^2$), smoking status (current, former, and never), physical activity (quintiles of MET hours/week), and total energy intake (quintiles of total caloric intake/day). An additional model was conducted without adding BMI as a covariate. Fixed-effects inverse variance weighted meta-analysis was used to combine cohort-specific results.

BMI, body mass index; CI, confidence interval; HR, hazard ratio.

There was no evidence of significant interactions on the multiplicative scale between diet quality and genetic risk on the risk of type 2 diabetes ($P_{interaction}$ = 0.65; Table 3). The lack of significant interactions was consistent when genetic risk was characterized using pathway-specific polygenic scores (Table 3) or when the DASH score was used (S6 Table).

In a joint analysis to investigate the combined association of genetic risk and diet quality with the risk of type 2 diabetes, there was a risk gradient with increasing genetic risk and decreasing diet quality (Fig 2). Age-adjusted estimates are presented in S6 Fig. Compared with individuals at low genetic risk and high diet quality, the multivariable-adjusted HR for risk of type 2 diabetes for low diet quality was 1.31 (95% CI 1.11, 1.54; *P* = 0.001) among those at low genetic risk, 1.53 (95% CI 1.31, 1.79; *P* < 0.001) among those at intermediate genetic risk, and 2.19 (95% CI 1.89, 2.54; *P* < 0.001) among those at high genetic risk. The joint association of diet quality and genetic risk was similar to the sum of the risk associated with each factor alone (RERI = 0.05, 95% CI −0.04, 0.13; $P_{interaction}$ = 0.30; Table 4), indicating no evidence of significant additive interactions. The proportion of contribution to excess type 2 diabetes risk was estimated to be 53.5% (95% CI 4.8, 62.2) to genetic risk, 38.6% (95% CI 29.4, 47.6) to diet quality, and 7.8% (95% CI −6.5, 22.2) to their interaction. We did not find evidence of additive interactions in crude models (S7 Table). We observed the same pattern for the joint associations and nonsignificant additive interactions when genetic risk was characterized using pathway-specific polygenic scores (S7 Fig). The proportion of contribution to excess type 2 diabetes risk due to genetic risk ranged from 61.2% (95% CI 51.9, 70.9) for the beta-cell dysfunction polygenic score to 21.9% (95% CI 5.9, 38.0) for the obesity-mediated insulin resistance polygenic score (Table 4). Findings from additive interaction analyses were similar when

**Table 3. Multiplicative interactions between diet quality and genetic risk on the risk of type 2 diabetes.**

| Polygenic score | Global polygenic score | Impaired insulin secretion | | Impaired insulin sensitivity | | |
|---|---|---|---|---|---|---|
| | | Beta-cell dysfunction | Impaired insulin synthesis | Obesity-mediated insulin resistance | Body fat distribution | Lipid/hepatic metabolism |
| **Multiplicative interaction** | | | | | | |
| Interaction term, coefficient | 0.99 (0.93, 1.05) | 0.94 (0.82, 1.00) | 0.98 (0.92, 1.04) | 1.02 (0.96, 1.09) | 0.97 (0.92, 1.03) | 0.96 (0.91, 1.02) |
| Interaction term, P value | 0.65 | 0.05 | 0.44 | 0.45 | 0.37 | 0.21 |

For each polygenic score, the combined interaction term coefficient and P value is shown. Estimates were obtained from Cox proportional hazards models with a cross-product interaction term between genetic risk and diet quality stratified by age and adjusted for ancestry-derived principal components, family history of diabetes, history of hypertension, history of hypercholesterolemia, menopausal status (women only), BMI, smoking status, physical activity, and total energy intake (methods). BMI, body mass index.

the 3 cohorts were analyzed separately (S8 Table, S8 Fig), when the DASH score was used (S9 Table, S9 and S10 Figs).

In a sensitivity analysis to investigate if BMI modified the joint association of increased genetic risk and low diet quality on type 2 diabetes risk, we showed that changes in BMI did not modify the risk of type 2 diabetes attributed to increased genetic risk and low diet quality ($P_{\text{interaction}} = 0.69$; S10 Table, S11 Fig).

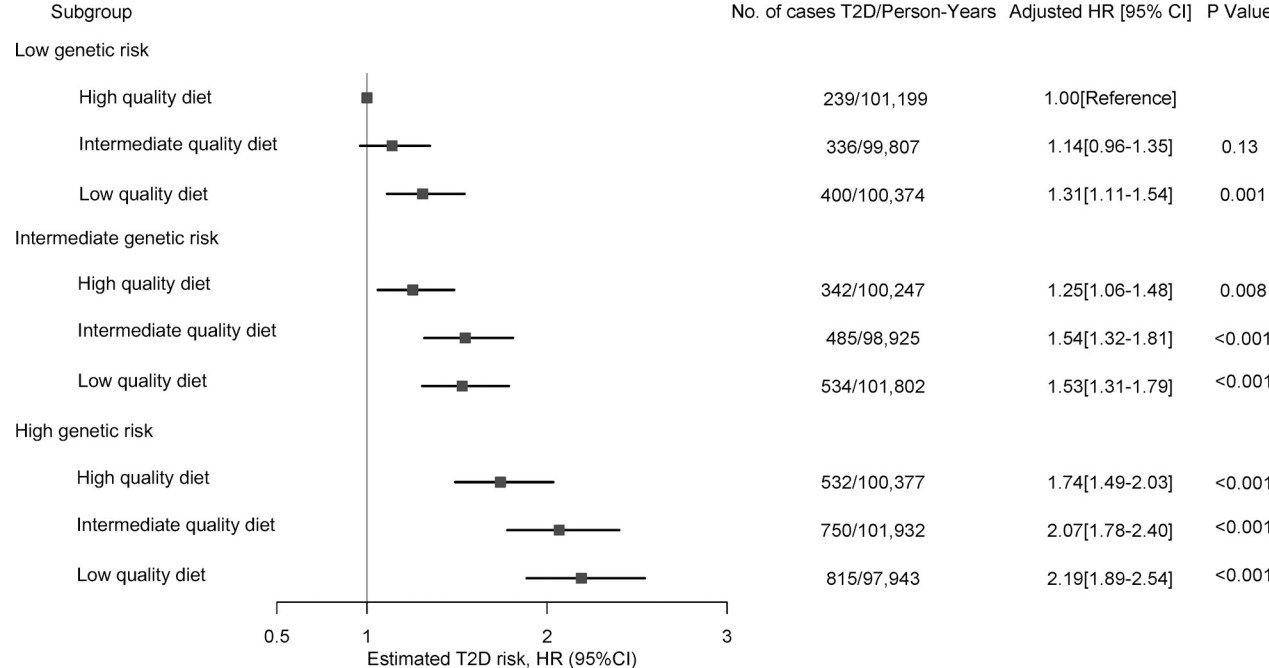

**Fig 2. Risk of incident type 2 diabetes according to categories of genetic risk and diet quality.** Shown are adjusted HRs and 95% CI of the estimate for type 2 diabetes in a pooled analysis of the 3 prospective cohorts according to categories of genetic risk and diet quality score. In these comparisons, participants with low genetic risk and high diet quality served as the reference group. Cox proportional hazards models were stratified by age and adjusted for ancestry-derived principal components, family history of diabetes, history of hypertension, history of hypercholesterolemia, menopausal status (women only), BMI, smoking status, physical activity, and total energy intake. Fixed-effects inverse variance weighted meta-analysis was used to combine cohort-specific results. BMI, body mass index; CI, confidence interval; HR, hazard ratio.

**Table 4. Additive interactions between diet quality and genetic risk using global and pathway-specific polygenic scores.**

| Polygenic score | Global polygenic score | Pathway-specific polygenic scores | | | | |
|---|---|---|---|---|---|---|
| | | Impaired insulin secretion | | Impaired insulin sensitivity | | |
| | | Beta-cell dysfunction | Impaired insulin synthesis | Obesity-mediated insulin resistance | Body fat distribution | Lipid/hepatic metabolism |
| **Main effects** | | | | | | |
| Diet quality[†] | 1.22 (1.14, 1.30) | 1.22 (1.15, 1.31) | 1.21 (1.14, 1.29) | 1.21 (1.13, 1.29) | 1.21 (1.14, 1.30) | 1.21 (1.14, 1.29) |
| Polygenic score[‡] | 1.29 (1.25, 1.33) | 1.26 (1.22, 1.30) | 1.14 (1.10, 1.17) | 1.09 (1.05, 1.12) | 1.23 (1.19, 1.26) | 1.11 (1.07, 1.16) |
| Joint effect | 1.57 (1.50, 1.64) | 1.51 (1.43, 1.58) | 1.36 (1.29, 1.44) | 1.32 (1.25, 1.40) | 1.48 (1.41, 1.56) | 1.32 (1.25, 1.39) |
| **RERI** | | | | | | |
| RERI | 0.05 (−0.04, 13.0) | 0.03 (−0.06, 0.11) | −0.01 (−0.08, 0.08) | 0.04 (−0.03, 0.12) | 0.03 (−0.07, 0.10) | −0.02 (−0.09, 0.05) |
| P value | 0.300 | 0.524 | 0.958 | 0.220 | 0.758 | 0.513 |
| **Attributable risk proportion, %** | | | | | | |
| Diet quality | 38.6 (29.7, 47.6) | 44.3 (34.9, 53.6) | 58.5 (46.7, 70.3) | 63.8 (50.8, 76.7) | 45.1 (35.4, 54.7) | 66.4 (53.2, 79.5) |
| Polygenic score | 53.5 (44.8, 62.2) | 61.2 (51.9, 70.5) | 42.1 (29.7, 54.4) | 21.9 (5.9, 38) | 52.2 (42.7, 61.8) | 41.4 (27.5, 55.3) |
| Additive interaction | 7.8 (−6.5, 22.2) | 5.4 (−11.9, 22.8) | −5.8 (−22.2, 21.1) | 14.2 (−7.1, 35.6) | 2.7 (−14.1, 19.4) | −7.8 (−32.1, 16.5) |

Multivariable-adjusted risk of type 2 diabetes estimated from Cox proportional hazards models stratified by age and adjusted for ancestry-derived principal components, family history of diabetes, history of hypertension, history of hypercholesterolemia, menopausal status (women only), BMI, smoking status, physical activity, and total energy intake.

Polygenic scores were standardized to allow comparisons across scores computed in this study with different the number of genetic variants. Details about the variants and weights used to compute theses scores are detailed in the Supporting information.

[†]Low quality diet versus high-quality diet was defined as a categorical variable based on the median distribution of the diet quality score in each cohort.

[‡]Per SD increase in polygenic scores.

BMI, body mass index; RERI, relative excess risk due to interaction; SD, standard deviation.

## Discussion

In this large prospective study to investigate how genetic risk and diet quality contribute to the risk of type 2 diabetes, we found that both low diet quality and increased genetic risk were independently associated with higher risk of type 2 diabetes without evidence of significant interactions. We showed that within any genetic risk category, high diet quality compared to low diet quality was associated with a nearly 30% lower risk of type 2 diabetes and that the risk of type 2 diabetes attributed to the combination of increased genetic risk and low diet quality was similar to the sum of the risks associated which each factor alone. Taken together, results from this study are important to support evidence-based prevention strategies for type 2 diabetes.

Our study adds to knowledge on the interplay between genetic and lifestyle factors by formally investigating whether polygenic scores for type 2 diabetes capturing overall genetic risk or distinct pathophysiological mechanisms could help prioritize individuals who would benefit the most from targeted dietary recommendations. Previous studies have shown no appreciable interactions between genetic and lifestyle factors on the development of type 2 diabetes [6,9,31], indicating that genetic risk does not modify the beneficial effect of healthy lifestyle interventions. However, previous studies considered a limited number of variants to generate type 2 diabetes polygenic scores, and the lack of significant interactions reported in these studies is often attributed to the mixture of variants affecting different pathways into a single score [6,32]. The latter is particularly relevant in the context of a highly heterogenous disease such as type 2 diabetes, in which groups of individuals are more likely to develop the disease due to

alterations in specific processes. By leveraging novel polygenic scores for type 2 diabetes, our study supports that both diet quality and overall or pathway-specific genetic risk are independently associated with risk of type 2 diabetes. These findings suggest that healthy dietary recommendations for the prevention of type 2 diabetes could be deployed across all levels of genetic risk in the population as genetic risk does not seem to modify their effectiveness. Further, our results emphasize the value of genetic risk assessment to identify individuals at increased disease risk and their potential for risk stratification and surveillance, as those at increased genetic risk might need to incorporate other lifestyle components in addition to healthy diet to mitigate their inherited risk.

We systematically evaluated the presence of additive and multiplicative interactions between genetic risk and diet quality on type 2 diabetes incidence. Interaction on a multiplicative scale means that the combined effect of 2 exposures is larger than the product of the individual effects of the 2 exposures, whereas interaction on an additive scale means that the combined effect of 2 exposures is larger than the sum of the individual effects [33]. While previous interaction studies have mainly tested for interactions on the multiplicative scale, the assessment of additive interactions is more suitable to identify which groups of individuals would benefit the most from a given intervention [34]. Our findings provide evidence that there is no departure from the additivity of risks attributed to each factor separately, indicating that the presence of the 2 exposures (low diet and increased genetic risk) does not explain a higher number of cases that could have prevented if only one of the exposures were present. These findings suggest that if interactions between genetic and dietary factors in type 2 diabetes exist, they are likely to be small, undetectable by conventional approaches, or influenced by other factors such as socioeconomic status or changes in body weight [35].

By clarifying that genetic risk and diet quality are each independently associated with the risk of type 2 diabetes and would not have an additive or multiplicative impact on the risk of the disease, our findings can yield useful clinical and public health answers as we prepare for the eventual implementation of precision nutrition. Major worldwide organizations recommend population-wide healthy dietary patterns for the prevention of metabolic diseases [36,37]. However, recent short-term multiomics feeding studies have reported large interindividual variability in response to specific foods or diets, supporting the need for more personalized approaches [38,39]. While long-term follow-up studies are needed to better appreciate the value of extremely personalized dietary recommendations for the prevention of diabetes and related metabolic diseases, findings from the present study support public health efforts that emphasize the consumption of healthy dietary patterns.

The strengths of this study include the use of new generation polygenic scores for type 2 diabetes that capture overall genetic risk or specific pathophysiological processes, the well-validated measures of dietary factors and the use of repeated diet measurements to reduce measurement error and noise, the large number of incident type 2 diabetes cases and extended follow-up, and the consistency of our findings in sensitivity analyses. Further, we generated both global and pathway specific polygenic scores to systematically investigate the presence of additive and multiplicative interactions between genetic risk and diet quality on the development of type 2 diabetes.

We acknowledge several limitations. Because this was an observational study and allocation to low- or high-quality diet was not randomized, we could not infer causality regarding the associations of low diet quality and increased genetic risk on the development of type 2 diabetes. A possible reason for the lack of interaction between the polygenic risk score and diet on the risk of type 2 diabetes could be imprecision in dietary intake measurement. We used cumulative averages of diet, which yield more precise dietary intake estimates than baseline

intakes alone [40], but the use of more objective dietary intake assessment methods, such as the use of smartphone applications, wearable technology, or dietary intake biomarkers, is necessary to accurately ascertain dietary intake and reduce self-reported errors [39,41]. We computed global and pathway-specific polygenic scores for type 2 diabetes, but the use of aggregated scores might have missed potential interactions driven by highly penetrant single genetic variants of strong effects or variants for glycemic traits interacting with environmental exposures [42]. However, further restraining the number of genetic variants will limit the clinical and public health value of our findings as highly penetrant variants tend to be rare or extremely rare in the population. In addition, the use of tails of polygenic score distribution (i.e., top 5% or 1% of genetic risk) could be used to detect potential interactions more likely to be present among people with very high or low genetic risk. However, such analysis would have lower statistical power compared to the assessment of interaction in the continuous scale, and it might yield spurious interactions due to unbalanced covariates between groups and residual confounding [43]. We restricted our analyses to participants for whom genetic data were available, which represents a small proportion of the original sample and might have induced selection bias. Participants for genetic determinations were selected to be representative of the original study population, and demographic characteristics and health status of participants with genetic information were generally similar to those who did not. The inclusion of well-informed and educated healthcare professionals without major chronic diseases at baseline might limit the generalizability of our findings to other populations. However, increased genetic risk and low diet quality have been associated with risk of type 2 diabetes in other populations [11].

In conclusion, our data provide evidence that genetic risk and diet quality are each independently associated with the risk of type 2 diabetes, without evidence of an additive or multiplicative impact on the risk of the disease. Our results suggest that the association of a healthy diet with lower risk of type 2 diabetes risk does not vary substantially based on the overall or pathway-specific genetic risk and highlights the potential of genetic risk assessment for future risk stratification and surveillance. Findings from this study might provide a valuable source of information for the primary prevention of type 2 diabetes.

## Supporting information

**S1 Fig. Distribution of polygenic scores in the NHS.** Distribution of the global and pathway polygenic scores in the NHS. NHS, Nurses' Health Study.
(PNG)

**S2 Fig. Distribution of polygenic scores in the HPFS.** Distribution of the global and pathway polygenic scores in the HPFS. HPFS, Health Professionals Follow-up Study.
(PNG)

**S3 Fig. Distribution of polygenic scores in the NHS II.** Distribution of the global and pathway polygenic scores in the NHS II. NHS, Nurses' Health Study.
(PNG)

**S4 Fig. Risk of incident type 2 diabetes associated with diet quality.** Shown are adjusted HRs and 95% CI of the estimate for type 2 diabetes in each of the 3 prospective cohorts per 10 units decrease in diet quality score assessed using the AHEI score. The diet quality score was derived from repeated measurements analyses. Cox proportional hazards models were stratified by age and adjusted for time-varying covariates including ancestry-derived principal components (not time-varying), family history of diabetes (not time-varying), history of hypertension, history of hypercholesterolemia, menopausal status (women only), BMI,

smoking status, physical activity, and total energy intake. Fixed-effects inverse variance weighted meta-analysis was used to combine cohort-specific results. The heterogeneity index ($I^2$) were used to assess heterogeneity. The $P$ values for the association were <0.001, <0.001, and 0.049 for the NHS, the HPFS, and the NHS II, respectively. AHEI, Alternate Healthy Eating Index; BMI, body mass index; CI, confidence interval; HPFS, Health Professionals Follow-up Study; HR, hazard ratio; NHS, Nurses' Health Study.
(TIFF)

**S5 Fig. Risk of incident type 2 diabetes associated with diet quality—sensitivity analysis using the DASH score.** Shown are adjusted HRs and 95% CI of the estimate for type 2 diabetes in each of the 3 prospective cohorts per 5 units decrease in diet quality score assessed using the DASH score. The diet quality score was derived from repeated measurements analyses. Cox proportional hazards models were stratified by age and adjusted for time-varying confounders including ancestry-derived principal components (not time-varying), family history of diabetes (not time-varying), history of hypertension, history of hypercholesterolemia, menopausal status (women only), BMI, smoking status, physical activity, total energy intake, and alcohol intake. Fixed-effects inverse variance weighted meta-analysis was used to combine cohort-specific results. The heterogeneity index ($I^2$) were used to assess heterogeneity. The $P$ values for the association were 0.001, <0.001, and 0.23 for the NHS, the HPFS, and the NHS II, respectively. BMI, body mass index; CI, confidence interval; DASH, Dietary Approaches to Stop Hypertension; HPFS, Health Professionals Follow-up Study; HR, hazard ratio; NHS, Nurses' Health Study.
(TIFF)

**S6 Fig. Risk of type 2 diabetes according to categories of the global polygenic scores and adherence to a healthy diet in age-adjusted secondary analyses.** Shown are age-adjusted HRs and 95% CI of the estimate for type 2 diabetes according to genetic risk and diet quality categories using the AHEI score. In these comparisons, participants with low genetic risk and high-quality diet served as the reference group. Cox proportional hazards models were stratified by age and adjusted for ancestry-derived principal components (not time-varying). A fixed-effects meta-analysis was used to combine cohort-specific results. AHEI, Alternate Healthy Eating Index; CI, confidence interval; HR, hazard ratio.
(PNG)

**S7 Fig. Risk of type 2 diabetes according to categories of the 5 pathway-specific polygenic score diet quality.** Shown are multivariable-adjusted HRs and 95% CI of the estimate for type 2 diabetes incidence according to pathway-specific polygenic score and diet quality categories. **(A)** Beta-cell polygenic score, **(B)** proinsulin polygenic score, **(C)** obesity polygenic score, **(D)** lipodystrophy polygenic score, and **(E)** liver metabolism polygenic score. In these comparisons, participants at low genetic risk with high-quality diet served as the reference group. A fixed-effects meta-analysis was used to combine cohort-specific results. CI, confidence interval; HR, hazard ratio.
(PDF)

**S8 Fig. Risk of incident type 2 diabetes according to genetic and diet quality risk in each cohort separately.** Shown are multivariable-adjusted HRs and 95% CI of the estimate for type 2 diabetes in **(A)** NHS, **(B)** HPFS, and **(C)** NHS II according to genetic risk and diet quality categories. In these comparisons, participants with low genetic risk and high-quality diet served as the reference group. CI, confidence interval; HR, hazard ratio; HPFS, Health Professionals Follow-up Study; NHS, Nurses' Health Study.
(PDF)

**S9 Fig. Risk of type 2 diabetes according to categories of the global polygenic scores and adherence to a healthy diet—sensitivity analysis using the DASH score.** Shown are multivariable-adjusted HRs and 95% CI of the estimate for type 2 diabetes according to genetic risk and diet quality categories using the DASH score. In these comparisons, participants with low genetic risk and high-quality diet served as the reference group. A fixed-effects meta-analysis was used to combine cohort-specific results. CI, confidence interval; DASH, Dietary Approaches to Stop Hypertension; HR, hazard ratio.
(PNG)

**S10 Fig. Risk of type 2 diabetes according to categories of the pathway specific polygenic scores and adherence to a healthy diet—sensitivity analysis using the DASH score.** Shown are multivariable-adjusted HRs and 95% CI of the estimate for type 2 diabetes incidence according to pathway-specific polygenic score and diet quality categories using the DASH score. **(A)** Beta-cell polygenic score, **(B)** proinsulin polygenic score, **(C)** obesity polygenic score, **(D)** lipodystrophy polygenic score, and **(E)** liver metabolism polygenic score. In these comparisons, participants at low genetic risk with high-quality diet served as the reference group. A fixed-effects meta-analysis was used to combine cohort-specific results. CI, confidence interval; DASH, Dietary Approaches to Stop Hypertension; HR, hazard ratio.
(PDF)

**S11 Fig. Interplay between diet quality and global polygenic score on type 2 diabetes risk according to changes in BMI.** Three-dimensional illustrations of type 2 diabetes risk, genetic susceptibility, and diet quality by BMI among individuals with normal weight **(A)**, overweight **(B)**, and obese **(C)**. The blue-colored region maps the lower risk area, and the red-colored area stands for higher risk area. Deciles of AHEI are inverse transformed, with 0 being good diet quality and 10 bad diet quality. Data from 3 cohorts were combined. Multivariate analyses were stratified by age and adjusted for time-varying confounders including cohort (not time-varying), ancestry-derived principal components (not time-varying), family history of diabetes (not time-varying), history of hypertension, history of hypercholesterolemia, menopausal status (women only), smoking status, physical activity, and total energy intake. $P = 0.681$ for 3-way interaction. AHEI, Alternate Healthy Eating Index; BMI, body mass index; SD, standard deviation; T2D, type 2 diabetes.
(PDF)

**S1 Table. Differences in baseline characteristics between the sample of participants included in this study and all participants in each original cohort.**
(DOCX)

**S2 Table. Characteristics of genetic variants used to build the 5 different pathway-specific polygenic scores.**
(DOCX)

**S3 Table. Associations of global and process-specific polygenic scores with type 2 diabetes risk in secondary analyses.**
(DOCX)

**S4 Table. Correlation between polygenic scores included in this study.**
(DOCX)

**S5 Table. Associations of global and process-specific polygenic scores with type 2 diabetes risk, random-effects meta-analysis.**
(DOCX)

**S6 Table. Multiplicative interactions between diet quality and genetic risk using global and pathway-specific polygenic scores.** Secondary analyses using the DASH score. DASH, Dietary Approaches to Stop Hypertension.
(DOCX)

**S7 Table. Additive interactions between diet quality and genetic susceptibility on type 2 diabetes risk, crude models.**
(DOCX)

**S8 Table. Additive interactions between diet quality and genetic susceptibility on type 2 diabetes risk in each cohort.**
(DOCX)

**S9 Table. Additive interactions between diet quality and genetic risk using global and pathway-specific polygenic scores.** Secondary analyses using the DASH score. DASH, Dietary Approaches to Stop Hypertension.
(DOCX)

**S10 Table. Interplay between diet quality and pathway-specific polygenic scores on type 2 diabetes risk by changes in BMI.** BMI, body mass index.
(DOCX)

**S1 Text. Type 2 diabetes polygenic scores.**
(DOCX)

**S2 Text. Prespecified analysis plan.**
(DOCX)

**S3 Text. STROBE checklist.** STROBE, Strengthening the Reporting of Observational Studies in Epidemiology.
(DOCX)

## Author Contributions

**Conceptualization:** Jordi Merino, Marta Guasch-Ferré, Jose C. Florez, Frank B. Hu.

**Data curation:** Marta Guasch-Ferré.

**Formal analysis:** Jordi Merino, Marta Guasch-Ferré, Jun Li, Wonil Chung, Yang Hu, Baoshan Ma, Jae H. Kang.

**Funding acquisition:** Peter Kraft, Qi Sun, JoAnn E. Manson, Walter C. Willet, Frank B. Hu.

**Investigation:** Jordi Merino, Jun Li.

**Methodology:** Jordi Merino.

**Visualization:** Yang Hu.

**Writing – original draft:** Jordi Merino, Marta Guasch-Ferré, Jun Li.

**Writing – review & editing:** Wonil Chung, Yang Hu, Baoshan Ma, Yanping Li, Jae H. Kang, Peter Kraft, Liming Liang, Qi Sun, Paul W. Franks, JoAnn E. Manson, Walter C. Willet, Jose C. Florez, Frank B. Hu.

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
