## [Editor Report · Decision Letter 0]

2 Apr 2021

Dear Dr Merino, 

Thank you for submitting your manuscript entitled "Polygenic scores, diet quality, and type 2 diabetes risk: a prospective, observational study" for consideration by PLOS Medicine.

Your manuscript has now been evaluated by the PLOS Medicine editorial staff and I am writing to let you know that we would like to send your submission out for external peer review.

Kind regards,

Caitlin Moyer, Ph.D.

Associate Editor

PLOS Medicine

---

## [Decision Letter · Decision Letter 1]

24 Aug 2021

Dear Dr. Merino,

Thank you very much for submitting your manuscript "Polygenic scores, diet quality, and type 2 diabetes risk: a prospective, observational study" (PMEDICINE-D-21-01465R1) for consideration at PLOS Medicine. 

Your paper was evaluated by a senior editor and discussed among all the editors here. It was also discussed with an academic editor with relevant expertise, and sent to four independent reviewers, including a statistical reviewer. The reviews are appended at the bottom of this email and any accompanying reviewer attachments can be seen via the link below:

[LINK]

In light of these reviews, I am afraid that we will not be able to accept the manuscript for publication in the journal in its current form, but we would like to consider a revised version that addresses the reviewers' and editors' comments. Obviously we cannot make any decision about publication until we have seen the revised manuscript and your response, and we plan to seek re-review by one or more of the reviewers. 

We expect to receive your revised manuscript by Sep 14 2021 11:59PM. Please email us (plosmedicine@plos.org) if you have any questions or concerns.

We look forward to receiving your revised manuscript. 

Sincerely,

Caitlin Moyer, Ph.D.

Associate Editor 

PLOS Medicine

plosmedicine.org

1. From the Academic Editor: Please emphasize the novelty, clinical significance and interpretation of the findings of the study, as compared to the existing literature describing relationships and interactions between lifestyle and genetic risk and risk of type 2 diabetes. Please provide some discussion on the new methods used in this study.

2. Data Availability statement: PLOS Medicine requires that the de-identified data underlying the specific results in a published article be made available, without restrictions on access, in a public repository or as Supporting Information at the time of article publication, provided it is legal and ethical to do so. Please see the policy at

http://journals.plos.org/plosmedicine/s/data-availability

and FAQs at

http://journals.plos.org/plosmedicine/s/data-availability#loc-faqs-for-data-policy

The Data Availability Statement (DAS) requires revision. For each data source used in your study:

3. Abstract: Please structure your abstract using the PLOS Medicine headings (Background, Methods and Findings, Conclusions).

4. Abstract Background: Provide the context of why the study is important. The final sentence should clearly state the study question.

5. Abstract: Methods and Findings: Please include some summary demographics of the cohorts, and specify the cohorts are US-based.. Please quantify the main results (with 95% CIs and p values). Please include the important dependent variables that are adjusted for in the analyses.

6. Abstract: Methods and Findings: In the last sentence of the Abstract Methods and Findings section, please describe the main limitation(s) of the study's methodology.

7. Author summary: At this stage, we ask that you include a short, non-technical Author Summary of your research to make findings accessible to a wide audience that includes both scientists and non-scientists. The Author Summary should immediately follow the Abstract in your revised manuscript. This text is subject to editorial change and should be distinct from the scientific abstract. Please see our author guidelines for more information: https://journals.plos.org/plosmedicine/s/revising-your-manuscript#loc-author-summary

8. Throughout: Please place in-text reference citations within square brackets.

9. Methods: Please include the information presented in the Appendix (Description of the study population and sample selection) where the Study design and Population are described.

10. Methods: Line 108: Please change the superscript reference to [10].

11. Methods: Diet quality: Please mention how the DASH score was determined. Please mention the food components for the AHEI and DASH.

12. Methods: Line 162: Please note where a complete list of the covariates/ risk factors may be found.

13. Methods: Line 202: Please change “duet” to “due” here.

14. Methods: Please ensure that the study is reported according to the STROBE guideline, and include the completed STROBE checklist as Supporting Information. Please add the following statement, or similar, to the Methods: "This study is reported as per the Strengthening the Reporting of Observational Studies in Epidemiology (STROBE) guideline (S1 Checklist)."

15. Methods: Did your study have a prospective protocol or analysis plan? Please state this (either way) early in the Methods section.

16. Results: In the first paragraph, please report on some of the participant characteristics presented in Table 1.

17. Results: Please provide 95% CIs and p values for all the main analyses described in the text (e.g PRS association with diabetes risk). For adusted analyses, please note the factors adjusted for (in the text or the appropriate figure/table), and please also provide results from unadjusted analyses.

18. Results: Line 271: Please mention the confounders adjusted for in this analysis.

19. Results: Line 275: Please clarify this sentence, as it seems as if there was no evidence for an interaction “...indicating a non-significant interaction on the multiplicative scale (P =0.65 for interaction…)”

20. Discussion: Line 288: Please rename this section “Discussion”

21. Discussion: Please present and organize the Discussion as follows: a short, clear summary of the article's findings; what the study adds to existing research and where and why the results may differ from previous research; strengths and limitations of the study; implications and next steps for research, clinical practice, and/or public policy; one-paragraph conclusion.

22. Please remove “Funding” and “Duality of Interests” sections from the main text. Please be sure that all information is completely and accurately entered in the Financial Disclosures and Competing Interests sections of the manuscript submission metadata.

23. References: Please use the "Vancouver" style for reference formatting, and see our website for other reference guidelines https://journals.plos.org/plosmedicine/s/submission-guidelines#loc-references

Please check journal title abbreviations- for example, please check if this should be J Womens Health for reference 15. Please update reference 31. Please use the same formatting for the references in the Appendix.

24. Figures and Tables: Please fully define all abbreviations used within each figure and table in the legend, including for the Supporting Information tables.

25. Figures 1 and 2: Please provide results from unadjusted analyses in addition to the adjusted analyses.

26. Table 2: Please provide the unadjusted analyses, and also please provide both 95% CIs and p values.

27. Supplementary Figure 7: Please place the axis labels so it is clear which axis is represented.

Comments from the reviewers:

Reviewer #1: This manuscript describes a detailed analysis of the potential for an interaction between diet and genetic risk for diabetes in regard to diabetes incidence. The authors conclude that any such interactions must be very small, and that a healthy diet is associated with lower risk of diabetes at all levels of genetic risk. 

Generally, the results are clearly presented, and the findings are robust. The findings are not surprising for two reasons. First, it is hard to see why high genetic risk would remove the effect of diet. Diet affects blood glucose levels even among people with T1 diabetes, so why would it not do so in people with a high polygenic risk for T2. Second, several other studies (as listed by the authors) have reported the same findings, although the current analysis takes a different and potentially more sophisticated approach.

The authors stopped updating dietary information after a diagnosis of CVD or cancer, and carried forward earlier cumulative averages. It is not clear why this reduces bias. If a person develops CVD, and then makes some dietary changes, it may be those exact dietary changes that explain why they didn't subsequently develop diabetes.

Abstract conclusion. 'gradients of genetic risk' is not quite correct. 'levels of genetic risk' would be better.

Table 1. The study legend refers to 'Age standardized characteristics', which may lead the reader to think that they are all standardised to the same age profile. However, the footnote indicates that each analytic sample has been standardised to the age profile of its own parent study. This is confusing. Furthermore, it's not clear what the purpose of this is, and I couldn't see where it was explained in the methods. The purpose of table 1 is to describe the precise population used in the current analysis, not to provide estimates of prevalences and means within the larger study from which the analytic sample has been drawn. Thus, it seems to me that there is no value in age-standardizing.

Figure 1. The labelling of the Impaired insulin sensitivity scores is a little confusing. For example, it looks as if score 3 is a genetic risk score for obesity. Also, I would number these 5 pathways 1-2 and 1-3. The current numbering of 1-5 obscures the fact that 3-5 come under the sub-heading of (b).

Titles for figures 1 and 2 use different terminology for what appears to be the same concept ('genetic susceptibility' vs 'genetic risk'). Please keep them consistent with each other.

Figure 2. It's not easy to tell from the legend how A and B differ from each other. For example, in 2A, the third row compares Low quality diet with High quality diet among people with low genetic risk. This seems to be the same comparison as presented in the first row of 2B. The HRs are the same (1.31), but the CIs and p-values are different.

Reviewer #2: The manuscript examines the association of genetic factors and diet quality with the incidence of type 2 diabetes. Novel insight is gained through systematic evaluation of additive and multiplicative interactions between genetic risk and diet quality, which were found to not be significant. Overall, the paper is very well-written and utilizes a number of secondary resources and metrics (e.g., DASH and AHEI scores) to ensure that results are robust and reproducible. The choice of statistical methods is appropriate, and the analysis of each cohort separately ensures that results are not confounded by cross-cohort batch effects. This is a well-executed study, and I only have one relatively minor suggestion.

In the appendix, the authors write that models were adjusted for the first 20 principal components (PCs). However, this raises a concern that if the dominant variance in the data aligns with whether individuals had type 2 diabetes, then such an adjustment would lower the signal-to-noise ratio. Furthermore, the choice of 20 appears to be arbitrary. It would be good to see that a) the top 20 PCs do in fact describe non-random variance in the data (e.g., via Horn's parallel analysis), and b) that the top 20 PCs do NOT have a significant association with type 2 diabetes.

Minor comments:

-The Introduction section is missing its caption

-To increase reproducibility and transparency, consider releasing R and SAS code as a publicly available repository (e.g., on GitHub).

-For completeness, please list the overall correlation between DASH and AHEI scores in each cohort.

-In Figure 2 caption, "..participants with low genetic risk and high-quality diet (Fig 2A) or high quality diet (Fig 2B) served as the reference group." seems unnecessarily redundant. Consider simplifying to "..participants with low genetic risk and high-quality diet served as the reference group."

Reviewer #3: Merino et al examined the joint effects of diet quality and genetic risk on incident T2D in three large prospective cohorts followed up for 902,386 person-years with available genetic data. This is a timely topic given the emerging concept of precision prevention with the implication that individuals with high genetic risk may benefit more from intervention. The authors leveraged the comprehensive genotype and phenotype data in UK Biobank to construct and validate a global polygenic score, as well as several pathway-specific polygenic scores for T2D, which were then analyzed jointly with the diet quality scores quantified by two different methods. The authors also conducted a three-way interaction analysis to investigate whether changes in BMI could modify the joint association of genetic risk and diet quality with T2D. The authors confirmed the independent associations of incident T2D with diet quality and genetic risk albeit without significant interaction and was not modified by BMI. They also quantified that 50% of the proportion of diabetes risk was explained by genetic risk and 38% by high quality diet. Within the polygenic score, ~60% was attributable to beta-cell dysfunction, although sub-analysis showed similar joint associations of quality diet and different pathway-specific polygenic risk scores. Overall, this is a well-conducted study in terms of design, sample size, cohort selection, exposure and outcome definition, and sensitivity analyses although some issues need to be addressed. 

Major issues:

One of the major limitations of this study is the applicability of the results to the general population since the majority of these subjects were well informed and educated healthcare professionals and predominantly nurses. Given the importance of education on diabetes risk, this point had not been discussed. In table 1, the author should state specifically the single sex nature in the 3 cohorts including 35,759 men and women. Although the authors concluded that high quality diet was associated with reduced risk of T2D irrespective of genetic burden, they proposed the use of polygenic score for surveillance purpose but did not discuss the implication of high quality diet within this context.

Background

1) Page 4, line 71-74: the authors listed four limitations in existing studies that prevented the researchers from identifying "genotypes interacting with lifestyle or dietary exposures" but did not address them in their analysis or discussion. 

* Limitation 1: Partial characterization of genetic risk due to a limited number of variants. There are genetic loci associated with other traits that may interact with diet or lifestyle on T2D risk (e.g. https://pubmed.ncbi.nlm.nih.gov/33864366/ reported gene-diet interactions that influenced HbA1c ). The "partial characterization of genetic risk" is not only restricted to "limited number of variants" but also the specificity, sensitivity and effect size of these scores. In Figure 1, the "Beta-cell dysfunction" score constructed from fewer SNPs compared with the "850,000 independent variants" used for the global polygenic score, yielded comparable hazard ratios on incident T2D. Thus, increasing the number of T2D risk loci does not necessarily provide additional variance in discriminating subjects with different T2D genetic risk. In the Appendix, the global polygenic score for T2D together with sex and age had an AUC of 0.638, what is the AUC when they are combined with the dietary scores in explaining the total variance? Given the differences in genetic architecture and many non-genetic factors across various populations and settings, emphasis on using aggregated polygenic risk score may impede our understanding in the effects of environment-personal-lifestyle-familial factors on the development of T2D.

* Limitation 2: Interactions in previous studies were predominantly assessed on the multiplicative scale alone. The authors only presented the additive Relative Excess Risk due to Interaction (RERI) in Table 2 when they should include the index estimates with confidence intervals and P values of multiplicative interactions for completeness.

* Limitation 3: Single time point dietary exposure. The cohorts had multiple diet quality scores collected at different time points. The authors should state clearly i) whether the diet quality scores were used as a time-fixed variable or a time-varying variable in the Cox model; ii) compared with the single time point diet quality score, e.g. the last one before endpoint, what was the advantage of the cumulative average? Was the cumulative average sufficient to capture the longitudinal trajectory of the score?

* Limitation 4: Limited follow-up. The three cohorts were followed up from 26 years to 30 years. With such long follow-up, considerable changes in the socioeconomic and demographic characteristics and ecological factors could influence on diet quality and T2D incidence. Did the author adjust for any socioeconomic covariates in their Cox regression analyses if there were relevant information in the questionnaire, or discuss the issue if not available. Did they adjust for year of data capture? 

Methods

2) For the global polygenic risk scores and pathway-specific risk scores, suggest 

* present the distribution of these PRS in the three cohorts separately;

* show overlap between the SNPs in the global and pathway-specific PRS;

* show correlation among the global PRS and pathway-specific PRSs 

* discuss AUC differences between UK Biobank validation dataset (0.723) and combined cohort under study (0.638)

3) Regarding the diet quality assessed at baseline and every 4 years thereafter, 

* What was the average number of diet quality scores for a subject in the three cohorts respectively? 

* Was there any correlation between the number of diet quality scores and cumulative average of the scores (i.e. did subjects with more diet quality scores have a higher cumulative average)? If so, this may be a marker of adherence and would that affect the interaction?

* What was the within-subject variance distribution in the three cohorts respectively? Is the cumulative average a valid surrogate of the long-term diet trajectory for subjects especially in those with fluctuating diet quality and large within-subject variance. 

* Was there any difference in the within-subject variance among the three cohorts? 

* The authors stopped updating dietary information when the subjects first reported having cardiovascular events or cancers. What about other factors that could dramatically change subjects' diet habits (e.g. T2D education or elevated glycemic traits that the subject may be aware of)?

* Did family history (FH) of diabetes affect the diet quality scores? Genetics is only a component of FH, did the authors repeat the analysis in those with or without FH? 

4) Page 6, line 183: The statement "Models were adjusted for time-varying …" is very vague. Please define the time-varying covariates and the "a priori" rationale for considering them as time-varying risk factors. Similarly, what were the time-fixed covariates in the models and the values at which time-point were used? A conceptual framework including these mediators or confounders would be helpful in explaining the selection of these covariates in the model. 

5) Page 6, line 185: The statement "The fully adjusted model was further adjusted …" is contradictory. 

6) Page 6, line 190-191: "Because time varying BMI could be on the causal pathway between diet quality and …, we also conducted separate models without adjusting for BMI". Where were the results of these separate models? In supplementary figure 7, the results are presented in 3 categories of BMI, was this the baseline BMI rathe than changes in BMI between baseline and censor point? 

7) Page 10, line 200: What was the "median distribution of the diet quality score in each cohort"? The statement is confusing from a statistical aspect.

8) Page 10, line 200: The RERI could also be calculated on a multiplicative scale to provide the index estimates of multiplicative interaction.

9) Page 10, line 205: "tested for multiplicative interactions by comparing the -2 log likelihood". The statement "-2 log likelihood" was informal. In addition, what test was used?

10) Page 10, line 209: The statement "examine whether changes in BMI modified the joint association …" is also vague. Which one was used to perform the 3-way interaction, time varying BMI or changes between two BMI? 

Results

11) In Table 1, please include the incidence of T2D, incidence of death, number of lost to follow up and number of cardiovascular/cancer events in the three cohorts.

12) Page 12, line 246: "there was a risk gradient with increasing …". Please perform trend analysis to demonstrate the statistical significance of such "risk gradient".

13) Page 12, line 251-252: "The available sample gave us 80% statistical power …", at what significance level?

14) Given that the three cohorts were all sex-specific (two women cohorts and one men cohort), did the authors observe any sex-specific association between the risk factors under study and T2D incidence?

15) The lack of interactions between T2D genetic risk scores and diet quality on incident T2D limited the novelty of this work. A recent study reported interaction between fruit intake and T2D genetic risk (https://pubmed.ncbi.nlm.nih.gov/33399975/). Did the author explore the association of a subset of foods or nutrients well known for their risk-conferring or risk-reducing effect for T2D instead of using the aggregated score .

16) Have the authors performed GWAS on dietary traits or borrowed similar information from the UK Biobank analysis (https://pubmed.ncbi.nlm.nih.gov/32193382/) to investigate whether there are shared genetic factors between T2D and unhealthy diet. Other similar work such as genome-wide gene-diet interaction analysis should preferably be performed or cited to reflect the limitation of the study (https://pubmed.ncbi.nlm.nih.gov/33864366/). These analyses would complement the existing results and strengthen this study.

17) Limitations and strengths of the study should be discussed.

Minor issues:

1) Page 4, line 82: "processes such impaired insulin …" -> "processes such as impaired insulin …"

2) Page 6, line 116: "… a global polygenic score for type 2 diabetes that capture overall …" -> "that captures overall …"

3) Page 7, line 138: "transcriptional regulation (13)' indicating …" -> "transcriptional regulation (13), indicating …"

4) Page 7, line 144-145: "in this study with different the number of …' -> "in this study with different number of"

5) Page 7, line 154: "we used the DASH score (Dietary Approaches to Stop Hypertension) …" -> "we used the Dietary Approaches to Stop Hypertension (DASH) score …"

6) Page 10, line 202: "which is the proportion of risk duet to …" -> "… due to …"

7) Page 26, line 566: "… or median (IQR) …" -> "… or medians (IQR) …"

8) Page 27, line 577: "time varying confounders including age, ancestry-derived principal components, family history …". How could the genetic principal components be used as time varying covariates? As for family history, can change in status over time influence dietary habit and thence risk of diabetes. 

Reviewer #4: The paper describes a prospective cohort study of interaction between genetic risk and diet quality on risk of type 2 diabetes. Three studies that followed up health professional's for over 900,000 person years were studied. Association is found with polygenic risk scores and with diet quality. These appear to additively affect risk of Type 2 diabetes with no evidence of interaction on the additive or multiplicative scale. 

This is a well done study and the work is well described. It provides further support for the idea that lifestyle modification is important independent of genetic risk. 

1. While I think the paper has useful findings, one concern I have is how much of an advance this is over the previous papers (referenced here from 5 to 11). It seems this lack of interaction between genetics and lifestyle has been well described several times in other studies, which is acknowledged in the paper. I think the main new finding claimed in this paper is the lack of additive interaction - but this seems like a limited advance over the previous studies. I would like more discussion - for a non-statistician - on the reasons why this is an important finding.

2. I also wonder why the UK Biobank study wasn't used in this study for comparison. That is also a longitudinal study and is used here to generate genetic risk scores, but I'm unsure why it also couldn't have been used for the main analysis - it would at least provide a comparison to a different population with different study selection criteria.

3. Diet quality is, by its nature, not a precise measure. The manuscript describes a good approach to assessing diet quality, but if you measure something imperfectly then this measurement error is going to make the possibility of finding interactions much harder. Do the authors think that this is a potential explanation for the lack of interactions identified?

4. The extreme of polygenic risk aren't assessed here (e.g.>5% or >1% tails of polygenic risk). While overall in the population there may be no obvious interactions, what about if someone has a very high genetic risk of diabetes? In these individuals you might imagine diet does have a less of an impact? Has this been tested?

[LINK]

---

## [Decision Letter · Decision Letter 2]

13 Dec 2021

Dear Dr. Merino,

Thank you very much for submitting your manuscript "Polygenic scores, diet quality, and type 2 diabetes risk: a prospective, observational study" (PMEDICINE-D-21-01465R2) for consideration at PLOS Medicine. 

Your revised paper was evaluated by a senior editor and discussed among all the editors here. It was also discussed with an academic editor with relevant expertise, and sent to four of the original reviewers, including a statistical reviewer. The reviews are appended at the bottom of this email and any accompanying reviewer attachments can be seen via the link below:

[LINK]

In light of these reviews, we will not be able to accept the manuscript for publication in the journal in its current form, but we would like to consider a revised version that addresses the reviewers' and editors' comments. Obviously we cannot make any decision about publication until we have seen the revised manuscript and your response, and we plan to seek re-review by one or more of the reviewers. 

We expect to receive your revised manuscript by Jan 03 2022 11:59PM. Please email us (plosmedicine@plos.org) if you have any questions or concerns.

We look forward to receiving your revised manuscript. 

Sincerely,

Caitlin Moyer, Ph.D.

Associate Editor

PLOS Medicine

plosmedicine.org

1. Please address the remaining points of reviewer 2, including the issue of including top principal components in the model, as well as acknowledging grant support and providing source code and all underlying data needed for supporting these analyses.

2. Title: We suggest removing the word prospective from the title. We suggest including the study setting/population in the title.

3. Data availability statement: Please revise the data statement, and please update this in the Data Availabilty section of the manuscript submission system. In your revision, you indicate: 

“Data Availability: Information including the procedures to obtain and access

data from the Nurses’ Health Study and Health Professionals Follow-Up Study is described at http://www.nurseshealthstudy.org/researchers for the Nurses’ Health Study (contact: nhsaccess@channing.harvard.edu) or https://www.hsph.harvard.edu/hpfs/hpfs_collaborators.htm for the Health Professionals Followup Study (contact: hpfs@hsph.harvard.edu). Access to statistical codes and datasets will be facilitated following the existing data sharing guidelines provided, which can be found on the study websites.”

PLOS Medicine requires that the de-identified data underlying the specific results in a published article be made available, without restrictions on access, in a public repository or as Supporting Information at the time of article publication, provided it is legal and ethical to do so. Please see the policy at 

http://journals.plos.org/plosmedicine/s/data-availability

and FAQs at 

http://journals.plos.org/plosmedicine/s/data-availability#loc-faqs-for-data-policy

 Please describe the locations of code and datasets necessary for replication of the analyses, or contact information for how these may be obtained. If the statistical codes and datasets are not freely available, please describe briefly the ethical, legal, or contractual restriction that prevents you from sharing those. Please also include an appropriate contact (web or email address) for inquiries (please note, this cannot be a study author).

4. Abstract: Background: At line 36, we suggest removing the word “prospectively” from the sentence.

5. Methods: Lines 128-131: Please provide either data or a reference supporting these sentences: “Participants for genetic determinations were selected to represent a representative sample of the original sample. Demographic characteristics and health status of participants with genetic information were generally similar to those who did not, therefore bias due to selection are probably minimized.”

6. Discussion: Line 419: Please temper this statement with “To the best of our knowledge, this is the first long-term…”

7. Page 25: Please remove the Author Contributions, Prior Presentation, and Data Availability sections from the main text, and please be sure all relevant information is entered in the manuscript submission system.

8. References: Please check the formatting of each reference, including journal title abbreviations. For example, reference 6 should be “PLoS Med” and reference 7 should be “Nat Commun”. Please use the "Vancouver" style for reference formatting, and see our website for other reference guidelines https://journals.plos.org/plosmedicine/s/submission-guidelines#loc-references

9. Figure 1 and Figure 2: Please also provide the unadjusted results. Please define all abbreviations in the legend (e.g. NHS, HPFS).

10. STROBE Checklist: Thank you for including the checklist. Please note “Funding Statement” as the location for item 22 “Funding” on the checklist.

11. S1 Appendix: Type 2 diabetes polygenic scores: Please define NHS I, NHS II and HPFS at first use in the text. Please also clarify “Genetic variants included in each polygenic score and corresponding weights for each variant are detailed in the appendix (pp).” Please also format references using “Vancouver” style, as in the main text.

12. S1 Table: In the legend, please define all abbreviations in the table.

13. S4 Table: Please define NHS I, NHS II and HPFS in the legend.

14. S9 Figure: Please define all abbreviations used in the figure in the legend.

Comments from the reviewers:

Reviewer #1: Thank you for addressing my comments.

Reviewer #2: Thank you for the opportunity to review the revised version of this work. Unfortunately, I feel that the authors have largely ignored my concerns in favor of working to address the more extensive criticism provided by the other reviewers.

In the response, the authors write that they followed the recommendations from investigators who developed BOLT-LMM. However, these recommendations are not quoted entirely accurately. Including principal components (PCs) as covariates to control for false positives is generally recommended by the BOLT-LMM developers for linear regression models, based on their earlier work [PMID: 16862161]. For mixed-model analysis, such as the one employed here, the original work describing BOLT-LMM states that "principal component analysis is not part of BOLT-LMM; it is unnecessary to perform PCA when running mixed model association methods" [PMID: 25642633]. Their later work does suggest "including PC covariates for the purpose of accelerating convergence of [...] BOLT-LMM" [PMID: 29892013], but algorithm runtime is of low importance in a one-shot analysis, especially when the source code is not being released for future reproducibility (see below). Blindly following a recommendation to improve runtime does nothing to address the original concern that the top principal components may have real (i.e., not ancestry-confounded false positive) type 2 diabetes signal, and that adjusting for these PCs may unnecessarily lower signal-to-noise ratio.

The URLs provided by the authors do NOT address my request to improve transparency and reproducibility of the study. The first URL is broken. The page at the second URL states that "Researchers using the NHS and NHSII data are required to acknowledge the grant support received by the National Institutes of Health (NIH) listed below, as appropriate, in all publications", yet the manuscript itself (which makes extensive use of these data) does not list any grants in the Acknowledgments section. Lastly, holding back source code under the guise of data availability is not appropriate. Data and software are two separate entities, and software licensing follows its own structure that is completely independent of any data usage agreements (https://en.wikipedia.org/wiki/Software_license). The authors need to decide what software license applies to their source code. If it falls under the umbrella of open source, then the code should be released as such. If the authors prefer to use a proprietary license, then it may be good to include a brief explanation why sacrificing transparency and reproducibility is appropriate in an NIH-funded study.

Reviewer #3: The authors have addressed all comments satisfactorily and the paper is much improved 

Reviewer #4: The authors have addressed my comments well.

[LINK]

---

## [Decision Letter · Decision Letter 3]

22 Feb 2022

Dear Dr. Merino,

Thank you very much for re-submitting your manuscript "Polygenic scores, diet quality, and type 2 diabetes risk: an observational study among 35,759 adults from three US cohorts" (PMEDICINE-D-21-01465R3) for review by PLOS Medicine.

I have discussed the paper with my colleagues and the academic editor and it was also seen again by one of the reviewers. I am pleased to say that provided the remaining editorial and production issues are dealt with we are planning to accept the paper for publication in the journal.

[LINK]

We look forward to receiving the revised manuscript by Mar 01 2022 11:59PM.   

Sincerely,

Caitlin Moyer, Ph.D.

Associate Editor 

PLOS Medicine

plosmedicine.org

Requests from Editors:

1. Reviewer 2 comments: As suggested, please complete the check to see how much T2D signal is present in the top PCs. Please also provide the relevant information needed to access the analysis code. 

2. Title: Please capitalize the first word of the subtitle, and please update this within the manuscript as well as the submission system: “Polygenic scores, diet quality, and type 2 diabetes risk: An observational study among 35,759 adults from three US cohorts”

3. Data availability statement: As mentioned by Reviewer 2, we request that you please make available the source code needed to replicate the study's findings in a repository (such as GitHub, SourceForge or Bitbucket) or a cloud computing service (such as Code Ocean). Protection of authors’ intellectual property will not be cause for exception. Please explain in the manuscript’s Data Availability Statement how readers can access the shared code and please also include an appropriate contact (web or email address) for inquiries (please note, this cannot be a study author).

4. Abstract: Line 40-41: Please revise if this should be: “Health Professional’s Follow-up Study”

5. Abstract: Line 53: We suggest revising to: “Limitations of this study include the self-report of diet information and possible bias resulting from inclusion of highly educated participants with available genetic data.”

6. Author summary: Line 63-64: Please revise to: “...the partial characterization of genetic risk and the predominant assessment of interactions on the multiplicative scale…” or similar.

7. Author summary: Line 79-81: Please clarify if this should be: “Further, we showed that the risk of type 2 diabetes attributed to the combination of increased genetic risk and low diet quality was similar to the sum of the risks associated with each factor alone.”

8. Methods: Line 116: Please change “RESEARCH DESIGN AND METHODS” to “Methods”

9. Methods: Line 252-253: Please describe how genetic risk categories (low, intermediate, and high) were established.

10. References: Please check journal title abbreviations (for example, reference 5 should be N Engl J Med).

11. Table 1: How was race/ethnicity defined and by whom? Should the DASH scores be reported here, in addition to the AHEI?

12. Table 3: The confidence interval for Beta-cell dysfunction is incomplete. 

13. Figure 1: Please note in the legend that IR is insulin resistance.

14. Figure 2: Please report p values as p<0.001 where relevant. Please report to two decimal places where p>/= 0.01 and please report to three decimal places where p<0.01.

15. Supporting Information File: Please provide a “clean” version of the document.

16. S1 Fig: Please increase the font size of the axis labels on the graphs, if possible.

17. S2 Fig and S3 Fig: Please provide p values for the associations for each of the three cohorts.

18. S4, S5, S6, S7, S8 Fig: Please report p values as p<0.001 where relevant. Please report to two decimal places where p>0.01 and please report to three decimal places where p<0.01.

19. STROBE Checklist: Please provide a “clean” version of the checklist.

Comments from Reviewers:

Reviewer #2: First of all, I apologize if I have been creating extra work for everybody involved. My original hope was that the authors would simply perform a quick check to see how much Type 2 Diabetes (T2D) signal is even present in the top principal components (PCs). Since PCs have to be computed for BOLT-LMM adjustment anyway, checking for correlation with T2D is a trivial amount of computation on top of that, and the very original comment could have been addressed with a single sentence to the effect of "To verify that adjusting for PCs did not remove a substantial amount of relevant signal, we examined stratification of patients into whether they have T2D by the top 20 PCs and found that <conclusion> (<relevant statistics here>)".

Instead, it appears that we are deep down the rabbit hole of discussing the merits of adjusting BOLT-LMM in general. To clarify, I agree with the authors that adjusting for the top PCs is a widespread practice, and that the resulting correction for possible confounders strongly outweighs any possible reduction in the signal of interest. I was simply suggesting to look at the amount of signal being lost (which was expected to be minimal), but this is a relatively minor sanity check that should not hold up the overall publication process. I also apologize if this didn't come across clearly in my earlier comments.

I can confirm that the financial disclosures statement correctly acknowledges relevant NIH grants. Unfortunately, the code is still being lumped under the data availability statement for some strange reason. I fully understand privacy concerns associated with releasing data, and the "available upon request" system is completely appropriate for it. However, computer code should not have the same constraints, and its release can help future researchers understand exactly what analyses were performed and with what parameters, thereby increasing overall reproducibility and robustness of the findings. It is a minimal amount of effort to upload existing analysis scripts to GitHub/GitLab or equivalent, add a simple README, and attach an open-source license (e.g., MIT or GPL). It is not entirely clear why the authors are so hesitant to publicly release their code, especially in an NIH-funded study, but this is not a hill that I am looking to die on with this manuscript.

Overall, I think this is a well-executed study, and the relatively minor points above should not prevent its progress towards a publication.

[LINK]

---

## [Editor Report · Decision Letter 4]

21 Mar 2022

Dear Dr Merino, 

On behalf of my colleagues and the Academic Editor, Weiping Jia, I am pleased to inform you that we have agreed to publish your manuscript "Polygenic scores, diet quality, and type 2 diabetes risk: an observational study among 35,759 adults from three US cohorts" (PMEDICINE-D-21-01465R4) in PLOS Medicine.

Please also address the following editorial requests:

1. Title: Please capitalize the first word of the subtitle, and please be sure to update this in the submission system: “Polygenic scores, diet quality, and type 2 diabetes risk: An observational study among 35,759 adults from three US cohorts”

2. Data availability statement: As discussed, please revise the Data Availability Statement to: 

“The data underlying the generation of the global polygenic score for type 2 diabetes are available from the UK Biobank project site, subject to registration and application process. Further details can be found at https://www.ukbiobank.ac.uk. This research was conducted under UK Biobank application no. 45052. Code to run the genome-wide association analysis for type 2 diabetes and generate the global polygenic score has been uploaded to GitHub (https://github.com/lab319/ps-diet-t2d).

Information including the procedures to obtain and access the data and codes used in this study in the Nurses’ Health Study I and II, and the Health Professionals Follow-Up Study is described at http://www.nurseshealthstudy.org/researchers for the Nurses’ Health Study (contact: nhsaccess@channing.harvard.edu) or https://www.hsph.harvard.edu/hpfs/ for the Health Professionals Follow-up Study (contact: hpfs@hsph.harvard.edu). The scripts to analyze NHS/HPFS data presented in this manuscript are open and widely available once access to the system is granted.”

Please ensure that all relevant scripts are available for access via the GitHub link provided.

3. Results: Line 318: Please check this sentence for a typo, and change to “When analyzed in each cohort separately, the risk of type 2 diabetes per 10-unit decrease in AHEI score was…” if appropriate.

4. Table 1 Legend (Page 30, Line 613): Please change “Caucasian” to “European” or another term that conveys what is meant.

PRESS

Sincerely, 

Caitlin Moyer, Ph.D. 

Associate Editor 

PLOS Medicine